# The Role of MicroRNAs in the Pathophysiology of Osteoarthritis

**DOI:** 10.3390/ijms25126352

**Published:** 2024-06-08

**Authors:** Dariusz Szala, Marta Kopańska, Julia Trojniak, Jarosław Jabłoński, Dorota Hanf-Osetek, Sławomir Snela, Izabela Zawlik

**Affiliations:** 1Orto Sport Center, 35-301 Rzeszow, Poland; daradasz@gmail.com; 2Department of Pathophysiology, Institute of Medical Sciences, Medical College of Rzeszow University, 35-959 Rzeszow, Poland; 3Student Research Club “Reh-Tech”, Medical College of Rzeszow University, 35-959 Rzeszow, Poland; juliatrojniak0@gmail.com; 4Faculty of Orthopaedic and Reumatology, Institute of Medical Sciences, Collegium Medicum, University of Rzeszow, 35-959 Rzeszow, Poland; jajablonski@ur.edu.pl (J.J.); dhanf@ur.edu.pl (D.H.-O.); ssnela@poczta.onet.pl (S.S.); 5Orthopaedics and Traumatology Clinic, Clinical Hospital No. 2, 35-301 Rzeszow, Poland; 6Department of General Genetics, Institute of Medical Sciences, Medical College of Rzeszow University, Kopisto 2a, 35-959 Rzeszow, Poland; izawlik@ur.edu.pl

**Keywords:** microRNA, osteoarthritis, OA, osteoarthrosis, miRNA, pathophysiology

## Abstract

Worldwide, osteoarthritis (OA) is the most common cause of joint pain in older people. Many factors contribute to osteoarthritis’ development and progression, including secondary osteoarthritis’ underlying causes. It is important to note that osteoarthritis affects all four tissues: cartilage, bone, joint capsule, and articular apparatus. An increasingly prominent area of research in osteoarthritis regulation is microRNAs (miRNAs), a small, single-stranded RNA molecule that controls gene expression in eukaryotes. We aimed to assess and summarize current knowledge about the mechanisms of the action of miRNAs and their clinical significance. Osteoarthritis (OA) is affected by the interaction between miRNAs and inflammatory processes, as well as cartilage metabolism. MiRNAs also influence cartilage cell apoptosis, contributing to the degradation of the cartilage in OA. Studies have shown that miRNAs may have both an inhibitory and promoting effect on osteoporosis progression through their influence on molecular mechanisms. By identifying these regulators, targeted treatments for osteoarthritis may be developed. In addition, microRNA may also serve as a biomarker for osteoarthritis. By using these biomarkers, the disease could be detected faster, and early intervention can be instituted to prevent mobility loss and slow deterioration.

## 1. Introduction

### 1.1. Osteoarthritis

Osteoarthritis (OA), also called osteoarthrosis, is the most common progressive chronic joint disease among older people. It leads to significant, chronic pain, the loss of mobility, and disability [1]. Osteoarthritis affects most people over the age of 65 [2]. An increasing trend in the number of people suffering from osteoarthritis has been observed around the world. An estimated 7% of the global population is affected by OA [3]. In 2019, there were over 500 million people with the disease [4]. The prevalence of OA is one in five among the general population and one in three among those over the age of 50 [3,5,6]. There are a variety of synovial joints that can develop OA, but the hands, knees, and hips are the most commonly affected [7]. There is an estimated 16% of the world’s population diagnosed with knee OA, with women suffering from the condition at a higher rate [6]. In the past, osteoarthritis was thought to be solely the result of cartilage “wear and tear”. Currently, it is understood as a complex, joint-wide condition involving matrix proteases and affecting the entire joint [8]. There are very limited non-surgical treatment options, which include physical activity plans, maintaining a healthy weight, and wearing appropriate footwear and supportive equipment [9]. In the advanced stages of the disease, patients who have not received long-term clinical treatment in the early stages of the disease may have to undergo joint replacement surgery. Globally, the number of these surgeries is on the rise by 10% each year, with 95% of them involving patients with osteoarthritis [10,11]. In the face of OA’s increasing incidence of hip and knee replacements, it is imperative to develop non-surgical treatments [12].

### 1.2. Risk Factors

A combination of cartilage and subchondral damage and other factors contribute to osteoarthritis pathophysiology. One of the key factors that contribute to the development of osteoarthritis is overweight and obesity [13,14,15,16,17,18,19]. Other factors include age >60 years and female gender [13,15,16,17,18,19,20,21,22,23]. Furthermore, mechanical factors contribute to osteoarthritis, such as professional works that place strain on the knee joints, injuries and previous joint surgeries, excessive exercise, running, and sedentary lifestyles [16,17,18,21,22,24,25,26,27,28,29]. Moreover, genetic mutations are also involved, as well as factors such as high bone mineral density (BMD) and diseases that impair deep sensation [17,21,24,29,30,31,32,33]—Table 1.

The main symptoms of osteoarthritis are joint pain, limited mobility, crepitations, and inflammatory changes of varying severity without systemic symptoms [34]. As a result of OA, the entire articular joint is affected by disorders [35]—Figure 1. A number of structural defects can be seen in articular cartilage, as well as bone loss in the sublymphatic region, as well as an increase in synovium and tissue hypertrophy. Ties and ligaments may also be unstable [10].

Two types of osteoarthritis exist: primary, which is more common and has an unknown cause; and secondary, which results from local damage to joints and abnormalities of the joint structure or systemic illnesses [24]—Table 2.

### 1.3. Clinical Relevance

MicroRNAs, also called miRNAs, are small single-stranded RNA molecules consisting of 19 to 25 nucleotides [38]. Human genes are estimated to be regulated by them to the extent of two-thirds [39]. By repressing translation or degrading target messenger RNAs, they are the key regulators of post-transcriptional gene expression [40,41,42,43]. By controlling cell differentiation, proliferation, apoptosis, and immune response, miRNAs can play an important role in many biological processes [44]. The dysregulation of miRNA expression, for example, resulting from mutations in the miRNA genes, may lead to serious disorders and contribute to the pathogenesis of many diseases, including osteoarthritis [45].

There is growing evidence that miRNA may play an important role in osteoarthritis pathogenesis [2,10,46,47,48].

In the context of osteoarthritis, miRNAs may affect various aspects of disease pathogenesis. As an example, some miRNAs play a role in modulating inflammatory processes that contribute to the development of OA and its progression. There is also the possibility that some miRNAs can influence cartilage metabolism by influencing the balance between anabolism and catabolism within cartilage cells. Moreover, some miRNAs can influence cartilage cell apoptosis, which contributes to OA cartilage degradation. Besides its effects on chondrocytes, miRNA also modulates osteoarthritis synovial fibroblasts (OASF) [49,50].

We aimed to summarize and evaluate the current understanding of the mechanism of action of miRNAs and their clinical significance. Understanding miRNAs’ role in OA pathophysiology may lead to new therapeutic targets and biomarkers. For example, if a specific miRNA is overexpressed in OA and contributes to cartilage degradation, it would be possible to develop a therapy that inhibits that miRNA to halt the progression of the disease.

## 2. Pathophysiology of Osteoarthritis

In recent years, the pathophysiology of osteoarthritis has been intensively studied. Given the complexity of the process, the initiation, development, and severity of osteoarthritis are determined by numerous factors. Although risk factors for the development of osteoarthritis have been identified, the molecular mechanisms that cause the development of this disease are not fully established [10].

The pathogenesis of osteoarthritis is a multi-aspect process that includes metabolic changes occurring in articular cartilage, subchondral bone, and synovial membrane. In the initial phase of osteoarthritis, increased porosity and reconstruction of the subchondral bone are observed [10]. Generally speaking, the development of osteoarthritis is initiated by disruption of the homeostasis of anti-inflammatory and pro-inflammatory factors. This condition is influenced by, among others, risk factors for osteoarthritis. As a result of many processes, cartilage degradation, bone remodeling and synovium proliferation occur [51]—Figure 2.

In osteoarthritis (OA), initial pathological changes manifest themselves first, mainly on the surface of the articular cartilage, especially in regions subjected to maximum load, where it is damaged [52,53]. Chondrocytes, which are the only cells in cartilage, exhibit increased proliferation in response to matrix degradation [54,55,56]. Some of these cells undergo a phenotypic transformation into hypertrophic chondrocytes, which resemble cells in the hypertrophic zones of the growth plate [57]. As the disease progresses, there is an intense degradation of the matrix caused by proteases that are induced by pro-inflammatory cytokines [54,58,59,60,61]. The pro-inflammatory cytokines involved in the pathogenesis of osteoarthritis include the following: IL-1β, TNF-α, IL-6, IL-15, IL-17, IL-18, IL-21, IL-22 [62,63,64,65,66,67,68,69,70,71]. These cytokines activate chondrocytes for the autocrine and paracrine production of further cytokines and proteases [1]. As a result, chondrocyte apoptosis and regions completely devoid of cells are observed in areas with significant matrix damage [59,72]. Osteoarthritis is also influenced by chemokines: CCL2, CCL3, CCl4, and CCL5, which contribute to pro-inflammatory cytokine production, and anti-inflammatory cytokines such as IL-4 and IL-10 [73,74,75,76,77,78]. The role of IL-4 and IL-10 is to suppress the progression of osteoarthritis [79]. Various cytokines, by activating numerous signaling pathways, lead to an increase in the expression of COX-2, which in turn results in an increase in the production of PGE-2. This phenomenon affects the breakdown of cartilage tissue and the formation of osteophytes. Although COX-2 inhibitors, such as nonsteroidal anti-inflammatory agents, are widely recommended as first-line treatment, they do not stop disease progression [80]. The focus on key cytokines influencing the development of osteoarthritis, such as IL-1β and TNF-α, and their suppression did not meet the expectations placed on them in the results of clinical trials [79,81,82,83,84]. In the case of OA, the action of individual cytokines may be independent, therefore blocking one of them is not always sufficient to effectively inhibit inflammation and the production of matrix-destroying enzymes [79].

As a result of many scientific studies, it has been found that subchondral bone sclerosis may be one of the main causes of osteoarthritis associated with aging [85]. In the place where the cartilage connects with the bone, a phenomenon has been observed where changes in the subchondral bone are inversely proportional to the degree of degeneration of the articular cartilage. It is noticed that the more the subchondral bone thickens, the more advanced the cartilage degeneration process is [1,86]. Moreover, abnormal bone remodeling processes resulting from disorders in the functioning of osteoblasts and osteoclasts play a key role in the initiation and progression of this disease [87,88,89,90]. A change in the subchondral bone plate occurs during both the early and advanced stages of osteoarthritis. In the early stages of the disease, the trabecular structure degrades, and bones become more porous, resulting in a larger distance between trabeculae and a decrease in bone mass [91]. As a result, the subchondral nuclear bone is more likely to deform under load. Bone loss is intensified as a result of microcracks that accumulate. In the advanced stage of the disease, the subchondral bone plate and trabeculae become thicker, but the bone loses mineralization and elasticity [91,92,93]. Moreover, bone cysts and osteophytes also form at the edges of joints, resulting in a flattened and deformed joint contour, which is known as bone abrasion [91,92,93,94].

Osteoarthritis is also accompanied by decreased bone density and mineralization, as well as irregularity in the structure of the bone matrix. These changes are believed to be induced by the transmission of signals from the costal cartilage through subchondral pores and by the process of vascular invasion. These changes occurring in bone and cartilage are considered crucial in the development of osteoarthritis [90]. Unfortunately, the exact mechanisms influencing the initiation and development of osteoarthritis through structural changes in bones and cartilage have not yet been fully explored. This is therefore the basis for conducting further, in-depth research in this area.

Research emerging in recent years indicates an important role of metabolism in maintaining the energy balance of articular chondrocytes. Metabolic dysfunctions of these cells are increasingly recognized as potential factors initiating and accelerating the development of osteoarthritis [95,96,97]. These metabolic changes affect metabolic pathways in chondrocytes, synoviocytes, and bone cells. The abnormally accelerated catabolism of articular chondrocytes, which promotes ECM degradation and preempts ECM synthesis, is a major feature of OA cartilage [10,98,99]. As a result of these processes, there is an interaction between these cells and the immune system, and the mediated regulation of inflammation can be considered a key element in understanding the pathogenesis of osteoarthritis [96].

Molecular pathways contributing to the development of osteoarthritis include the increased activity of extracellular matrix (ECM)-degrading enzymes, such as *matrix metalloproteinase 13 (MMP-13)* and ADAMTS-5, as well as the increased production of inflammatory cytokines, including interleukin-1β (IL-1β), IL-6, and tumor necrosis factor (TNF). Additionally, apoptotic pathways are activated, including caspase 3 and poly (ADP-ribose) polymerase (PARP), with a simultaneous reduction in the expression of genes responsible for the synthesis of ECM components, such as *COL2A1* and *ACAN*, and changes in processes maintaining cellular balance, including autophagy [100,101,102,103].

## 3. The Role of miRNAs in the Pathogenesis of Osteoarthritis

The currently published results indicate the high importance of miRNAs in the pathogenesis of osteoarthritis. Data indicate increased concentrations of most miRNAs in patients with osteoarthritis, and therefore, in most cases, the increased promotion of osteoarthritis by various potential targets—Table 3. The reports are mostly innovative and indicate more and more miRs, the increased concentrations of which are observed in osteoarthritis.

There are many potential mechanisms by which microRNAs promote or prevent osteoarthritis.

By far the best-studied microRNA in cartilage is miR-140 [104]. Studies have shown that miR-140-5p is crucial in directly regulating the levels of genes including, but not limited to: *IGFBP-5*, *MMP13*, *Hdac4*, *Cxcl12*, *Bmp2*, and *SMAD3*. These genes play an important role in the development of chondrocytes and in maintaining the balance of joint cartilage [105,106,107]. However, it appears that miR-140-3p is most abundantly expressed in cartilage, and perhaps miR-140-5p is most important in the development of osteoarthritis, while miR-140-3p potentially plays a greater role in the homeostasis of joint tissues [108,109]. It was found that the level of miR-140 in the knee cartilage of people suffering from osteoarthritis is lower compared to healthy cartilage. The complete elimination of miR-140 in mice resulted in a mild form of dwarfism and impaired chondrocyte differentiation and proliferation [110,111]. Additionally, the complete removal of miR-140 increased the tendency of mice to spontaneous age-related cartilage degeneration and to intensify cartilage damage in the case of surgically induced osteoarthritis [111]. The deletion of miR-140 combined with the inhibition of the let-7 microRNA in mice in turn leads to more severe changes in skeletal structure than were observed with any single mutation [112].

MiR-146a is another microRNA that has been extensively studied and is known to be stimulated by a variety of inflammatory factors. It plays an important role in the functioning of the immune system and inflammatory processes [113]. miR-146a is highly expressed in the cartilage tissue of people suffering from early-stage osteoarthritis and has the potential to regulate and modulate pain associated with this disease [114,115]. It has been reported that it may affect inflammatory processes, autophagy, and apoptosis mechanisms in cartilage cells, as well as the activity of genes responsible for the composition of the extracellular matrix [116,117,118]. In their study, Zhang et al. analyzed miR-146a-5p and found that it was highly active in the knee cartilage of individuals with osteoarthritis (OA) [119]. Conversely, the NUMB protein exhibited low activity and was suppressed by miR-146a-5p. The rise in miR-146a-5p levels may lead to enhanced cellular apoptosis and reduced autophagy in human and mouse chondrocytes by affecting factors such as active caspase-3, PARP, Bax, Beclin 1, *ATG5*, p62, LC3-I, and LC3-II. Elevating the level of weakly expressed *NUMB* could neutralize the effects of miR-146a-5p on chondrocyte apoptosis and autophagy. Furthermore, the injection of miR-146a-5p antagomir directly into the joint could reverse the effects of miR-146a-5p on apoptosis and autophagy in chondrocytes in OA mice. As a result of these studies, it was shown that reducing miR-146a-5p activity inhibited apoptosis and supported autophagy in chondrocytes, targeting *NUMB* in both in vivo and in vitro studies [119]. These findings are also supported by another study using miR-146a-5p. Qin et al. demonstrated that knockdown of miR-146a-5p in chondrocytes antagonizes IL-1β-mediated inflammatory responses and increases catabolism in vitro and attenuates cartilage degeneration in injury-induced OA in mice [120].

However, different results were obtained by Guan et al., where they found reduced miR-146a expression in areas affected by osteoarthritis (OA) compared to healthy cartilage. These researchers also described mice genetically lacking miR-146a that showed spontaneous OA symptoms early in the disease development, while mice overexpressing miR-146a in chondrocytes were resistant to OA. Additionally, mice lacking miR-146a were more sensitive to joint instability-induced OA, whereas mice with controlled overexpression of miR-146a were protected from the disease. It appears that miR-146a can protect against OA by affecting the Notch1 protein, and the delivery of Notch1 inhibitors to the joints of mice lacking miR-146a prevented joint destruction [121]. However, genes targeting miR-146a-3p currently have less evidence to correlate with osteoarthritis [122]. Although studies on the single nucleotide polymorphism (sNP) of miR-146a did not confirm an increased risk of osteoarthritis (OA) as a result of the mutation, they did indicate a decrease in the expression level of miR-146a caused by the mutation, which in turn led to an increase in the activity of the *IRAK1* and *TRAF6* genes [123]. These findings, while promising, require further research to clarify the contradictions.

In a study that detected the presence of miR-9-5p in patients with osteoarthritis, it was shown that this miR promoted cell proliferation and suppressed chondrocyte apoptosis by affecting, among other things, *matrix metalloproteinase-13 (MMP-13)*, which is the primary MMP involved in in the degradation of cartilage through its special ability to cleave type II collagen [124,125]. The second target turned out to be protogenin *(PRTG)*, the overexpression of which induces the activation of caspase-3 signaling and increases apoptosis. Since in OA patients, the expression of *PRTG* negatively correlated with high miR-9 expression and therefore turned out to be reduced, it did not show an apoptotic effect [124,126]. It turns out that this selected microRNA can reduce the progression of osteoarthritis.

Another microRNA present in patients with osteoarthritis turned out to be miR-10a-5p. Its specific target is *HOXA3*, the silencing of which significantly inhibited chondrocyte proliferation, and promoted chondrocyte apoptosis and cartilage matrix degradation [127].

Another one, miR-22, activates metalloproteinases and aggrecanases and downregulates the structural proteins of cartilage, which leads to its degradation. PPARα and BMP-7 are potential targets [128]. PPARs are ligand-activated receptors in the nuclear hormone receptor family. The human gene for PPARα is located on chromosome 22. The activation of PPARγ and PPARα has been shown to effectively modulate the NF-κB, AP-1, and other stress-responsive oxidative signaling channels, leading to the inhibition of inflammatory responses. Furthermore, the activation of PPARγ and PPARα may provide protection to chondrocytes by exerting control over their autophagic behavior [129]. The overexpression of miR-22 inhibited BMP-7 and PPARα protein expression, helping to promote osteoarthritis. Moreover, miR-22 expression was positively correlated with BMI [128].

As numerous studies have shown, miR-27b is also involved in promoting osteoarthritis by targeting, among others, *MMP-13*, *COL1A1*, and ADAMTS8. Metalloproteinase with thrombospondin motifs (ADAMTS) causes the degradation of extracellular matrix (ECM) collagen and aggrecan II [130]. ADAMTS is also one of the targets of miR-34a-5p, miR-140, and miR-140-5p [114,131,132].

The findings also indicate that OA synovial fibroblasts (OASFs) are characterized by increased levels of proinflammatory cytokines relative to normal synovial fibroblasts (NSF). Moreover, it was noticed that miR-149-5p, which was decreased in patients with osteoarthritis compared to the control group, plays a role in reducing the expression level of IL-1β, IL-6, and TNF-α [133]. Thus, significantly reduced levels of miR-149-5p interfere with the synthesis of connective protein and proteoglycan [133,134]. Furthermore, Jiang et al. showed that increased miR-149 or suppressed vascular cell adhesion molecule 1 (VCAM-1) reduced inflammation and apoptosis in the cartilage tissues of OA mice, which was associated with the inactivation of the PI3K/AKT pathway [135]. In addition, miR-149 suppressed the chondrocyte inflammatory response that was induced by IL-1β by downregulating the activation of TAKI/NF-κB signaling pathway also counteracted osteoarthritis [136]. The therapeutic effect of the anti-inflammatory drugs used in the study was related to their ability to suppress the expression of miR-149-5p, suggesting that the regulation of these miRNAs may constitute an innovative method of reducing inflammation in OA.

Studies have shown that miR-128 is also involved in the pathogenesis of osteoarthritis [137]. According to Lian et al., miR-128a inhibits the autophagy process in cartilage cells and worsens the symptoms of osteoarthritis in the knee joint by affecting Atg12 [137]. Mice with the miR-128a gene deleted presented less abnormalities in microcomputer tomographic and kinematic measurements after DMM surgery and showed less advanced changes in cartilage loss at the histological level [138].

MiR-210 acts as a positive regulator of osteoblastic differentiation by inhibiting the TGF-β/activin signaling pathway through the inhibition of AcvR1b [139]. Additionally, studies have shown that miR-210 acts as an inhibitor of the production of pro-inflammatory cytokines [140]. Recent reports indicate that miR-210 is associated with the NF-κB signaling pathway, which plays an important role in regulating the immune response, inflammatory processes, and cell survival, and therefore has a protective effect against the development of osteoarthritis [140,141].

Data also report a role for miR-335-5p in inhibiting osteogenic and adipogenic differentiation and promoting extracellular matrix (ECM) degradation. The Wnt and IFNγ signaling pathways and the *HBP1* gene have been indicated as targets of miR-335-5p [124,142]. The obtained data also indicate the activation of the NF-κB pathway and a significant increase in the levels of IL-1α and IL-6 in cells after transfection with miR-335-5p mimics, compared to control cells [142].

Finally, miR-485-5p, which is overexpressed in osteoarthritis, can downregulate *SOX9*. By doing so, it can inhibit the differentiation of BMSCs into chondroblasts and promote the expression of inflammatory factors to accelerate the development of osteoarthritis [143].

**Table 3 ijms-25-06352-t003:** The role of selected microRNAs in the osteoarthritis process—potential effect, impact on potential target genes/pathways, and attitude towards osteoarthritis.

MicroRNA	Level in OA	Potentially Action	Potential Target	Role in OA Pathogenesis	References
miR-9-5p	↓ ^1^	intensifying proliferation and suppressing chondrocyte apoptosis	*MMP-13*, PRTG	− ^3^	[124]
miR-10a-5p	↑ ^2^	inhibiting chondrocyte proliferation, promoting chondrocyte apoptosis, and promoting cartilage matrix degradation	*HOXA3*	+ ^4^	[127]
miR-22	↑	the activation of metalloproteinases and aggrecanases and downregulation of cartilage structural proteins, cartilage degradation	PPARα, *BMP-7*	+	[128]
miR-27b	↑	the fibrosis of the synovial membrane, influence on inflammatory processes, cartilage metabolism, and apoptosis of cartilage cells	*MMP-13*, *COL1A1*, α-SMA2, ADAMTS8, and *CBFB*	+	[144,145,146]
miR-34a-5p	↑	cell cycle arrest, promoting apoptosis, senescence, and proliferation	*COL2A1*, *ACAN*, *ATG5*, *MMP13*, ADAMTS5, IL-1β, and *COL10A1*	+	[131]
miR-127-5p	↓	increasing the synthesis of cartilage extracellular matrix (ECM)	Osteoponin and *MMP-13*	−	[147,148]
miR-128a	↑	impaired chondrocyte autophagy, the suppression of extracellular matrix deposition	*Atg12*, *Bax*, *Bcl2*, and cleaved caspase-3	+	[137,138]
miR-138-5p	↑	the degradation of cartilage extracellular matrix (ECM)	*FOXC1* and increase in IL-1β	+	[124]
miR-140	↓	promoting chondrocyte differentiation	ADAMTS5 and AGGRECAN	−	[132]
miR-140-3p	↓	increase in the viability and migration capacity of chondrocytes	increase: *SOX-9*, *COL2*, *ACAN*, *RUNX2*, and *SCX*, decrease: *COL1*, *COL6*, *COMP*, *TNC*, and *FMOD*	−	[149]
miR-140-5p	↓	inhibits inflammation in the joint cavity, inhibits the progression of OA, promotes chondrogenesis, inhibits chondrocyte apoptosis, inhibits chondrocyte hypertrophy	IGFBP-5, IL-1β, IL-6, Syndecan-4, ADAMTS5, *MMP-13*, *SMAD3*, *HMGB1*, *RALA*, *FUT1*, HDAC4, and SMAD1	−	[105,109,111,150,151,152,153,154,155,156]
mi-146	↑	promoting the inflammatory response in the joint	*TRAF6* and *IRAQ1*,	+	[157,158]
miR-146a-5p	↑	cartilage degradation, synovitis, neoangiogenesis, and osteoclastogenesis	TNF α, IL-1β, *TRAF6* and *IRAK1* genes, and *MMP-13*	+	[114,124,159]
miR-149	↓	promoting the synthesis of connective protein and proteoglycan, suppressing the inflammatory process	TNFα, IL1β, IL6, VCAM-1, and *TAK1*	−	[134,135]
miR-210	↓	promoting osteoblastic differentiation, anti-apoptotic effect, anti-inflammatory effect	AcvR1b and DR6	−	[139,140,141,160]
miR-335-5p	↑	osteogenic and adipogenic differentiation, promoting ECM degradation	Wnt signaling pathway, IFNγ, HBP1, *ACAN*, *MMP13*, collagen X, and collagen II	+	[124,142]
miR-485-5p	↑	inhibiting the differentiation of BMSCs into chondroblasts and promoting the expression of inflammatory factors	*SOX9*	+	[143]

^1^ ↓—downregulation. ^2^ ↑—upregulation. ^3^ −—impairment. ^4^ +—intensification.

These results indicate a complex role of miRNAs in the regulation of pathological processes in OA, including cartilage degradation, inflammatory processes, cell differentiation, and tissue homeostasis. The most-studied miRNAs in cartilage are miR-140 and miR-146-5p.

Most miRNAs show increased levels in OA patients, which may contribute to its progression through various biological targets. However, changes in miRNA expression are also observed in osteoarthritis, some of which are downregulated, such as miR-127-5p and miR-140-5p. The reduced expression of the mentioned miRs correlates with the formation of changes in the joints and thus promotes osteoarthritis. However, miR-127-5p and miR-140-5p themselves have a protective effect on cartilage, promoting the differentiation of chondrocytes, preventing the formation of a planar state, and increasing the synthesis of the extracellular matrix. Changes in the expression of these miRNAs may therefore constitute potential targets for new therapies in the treatment of OA.

According to the analysis, some miRNAs are upregulated in OA, such as miR-146a-5p and miR-34a-5p, while others are downregulated, such as miR-140-5p. The osteoarthritis-related miRNA-22, for instance, induces inflammation and catabolism in joint cells. Other miRNAs, such as miRNA-9 and miRNA-98, can inhibit the secretion of *matrix metalloproteinase-13 (MMP-13)* and the inflammatory factors TNF and IL1β, suggesting their potential role in inhibiting cartilage degradation.

## 4. Potential Therapeutic Targets

Studies have shown that miRNAs may play a key role in the pathogenesis of osteoarthritis by regulating the expression of genes related to articular cartilage homeostasis and inflammatory processes. For example, miR-146a-5p and miR-34a-5p are upregulated in OA and may serve as diagnostic biomarkers and potential therapeutic targets. On the other hand, miR-127-5p, and miR-140-5p are downregulated and may have a protective function in maintaining cartilage integrity [161].

The use of miRNAs in OA therapy may include strategies to modulate their expression to restore articular cartilage homeostasis. For example, the delivery of synthetic miRNA mimetics or inhibitors may help restore the normal miRNA expression profile in diseased joints. Additionally, miRNA therapy can be combined with other approaches, such as cell therapy using mesenchymal stem cells, which are also regulated by miRNAs [161].

However, there are challenges associated with miRNA therapy, including the specificity of delivery, stability of miRNA molecules in the body, and potential adverse effects resulting from the modulation of multiple gene expression [162]. MiRNAs are characterized by a short lifespan, limited stability in the body, problems with distribution in tissues, interference with natural RNA processes, and the possibility of causing undesirable effects [163]. Further research is necessary to understand the complex interactions of miRNAs with their target genes and to develop effective and safe methods for miRNA delivery to joint tissues.

According to some studies, osteoarthritis patients have a much lower expression of certain miRNAs, such as miRNA-27b, than the patients without the disease [29]. It has also been found that miRNA-27b regulates MMP-13 expression in human chondrocytes, indicating that miRNAs may be used therapeutically for the treatment of osteoarthritis. Understanding the interactions between miRNAs and their multiple target genes may be crucial for regulating homeostasis and control pathways in osteoarthritis. As such, these miRNAs may serve as biomarkers for osteoarthritis early diagnosis.

In summary, miRNAs present promising potential as therapeutic targets in OA, offering new opportunities for the treatment of this disease. The development of miRNA therapy may contribute to improving the quality of life of OA patients, but this requires further research and development of miRNA delivery technologies. Understanding the role of miRNAs in the pathophysiology of osteoarthritis may lead to the identification of new therapeutic targets and biomarkers of the disease. For example, if a specific miRNA is overexpressed in OA and contributes to cartilage degradation, it would be possible to develop a therapy that inhibits that miRNA to halt the progression of the disease. Alternatively, if a specific miRNA is deficient in OA and its absence contributes to disease progression, it could be possible to develop a therapy that increases the expression of that miRNA to inhibit disease progression.

## 5. Discussion

MiRNAs play a critical role in osteoarthritis (OA), influencing cartilage degradation, inflammation, cell differentiation, and tissue homeostasis. Patients with OA generally have increased miRNA concentrations, which generally promote the disease through various biological mechanisms.

One of the most extensively studied miRNAs in cartilage is miR-140. As a direct regulator of chondrocyte development and cartilage balance, MiR-140-5p is known to regulate genes such as *IGFBP-5*, *MMP13*, *HDAC4*, *CXCL12*, *BMP2*, and *Smad3* [105,106,107]. MiR-140-3p is abundant in cartilage, maintaining joint tissue homeostasis, while miR-140-5p is implicated in OA development [111]. It has been shown that miR-140 levels are diminished in OA cartilage, resulting in impaired chondrocyte differentiation and proliferation, and increased cartilage degeneration [105,111].

OA cartilage expresses high levels of MiR-146a, which targets the NUMB protein to modulate inflammation, autophagy, and apoptosis in chondrocytes [119]. By suppressing MiR-146a-5p, chondrocytes are able to mitigate apoptosis and enhance autophagy [119,164]. OA symptoms are linked to miR-146a downregulation, while overexpression provides resistance to OA by affecting the Notch1 protein [121]. In light of miR-146a’s complex role in OA, further research is needed to resolve contradictory findings.

OA pathogenesis is also influenced by other miRNAs. Through its targeting of MMP-13 and PRTG, MiR-9-5p promotes chondrocyte proliferation and inhibits apoptosis [124]. Through targeting *HOXA3*, MiR-10a-5p inhibits chondrocyte proliferation and promotes apoptosis and cartilage degradation [127]. By activating metalloproteinases and aggrecanases, MiR-22 downregulates structural proteins in cartilage and targets PPARα and BMP-7 [128]. The miR-27b promotes osteoarthritis by targeting MMP-13, COL1A1, and ADAMTS8, which results in the degradation of the ECM [130]. It is reported that miR-34a-5p induces apoptosis and senescence by interacting with several genes, including *MMP13* and IL-1β [131].

MiR-149-5p, reduced in OA, plays a role in decreasing IL-1β, IL-6, and TNF-α expression, with potential anti-inflammatory effects [133]. MiR-128a inhibits autophagy in cartilage cells, aggravating OA symptoms [137]. MiR-210 inhibits pro-inflammatory cytokine production and regulates osteoblastic differentiation [140,141]. There has been evidence that MiR-335-5p promotes the degradation of ECM and may be associated with the NF-B pathway activation, increasing IL-1α and IL-6 levels [142]. MiR-485-5p, overexpressed in OA, inhibits chondroblast differentiation by downregulating SOX9 [143].

There is a complex regulatory network in the body in which one miRNA can influence multiple molecular targets at the same time, while multiple miRNAs can affect the same target at the same time. The same miRNAs may be responsible for regulating different molecular pathways and pathogenic mechanisms in osteoarthritis. For example, there are several miRNAs that influence IL-1β levels, including miR-140-5p, miR-149-5p, and miR-146a-5p. A similar situation is in the case of IL-6, which is acted upon by miR-140-5p and miR-149-5p. *MMP13* is also a target of several miRNAs. It is a target of, among others, miR-335-5p, miR-9, miR-98, miR-27b, miR-34a-5p, miR-140, and miR-140-5p. In some cases, miRNAs can regulate the same molecular target together. This is known as cooperative miRNA interaction, which may lead to a synergistic effect in regulating osteoarthritis-related gene expression [165]. There may be diversity in gene regulation by miRNAs in individuals with osteoarthritis, resulting in differential gene expression and genetic variants [166]. Because of this, individual genetic and epigenetic characteristics may determine which miRNAs regulate the same molecular targets.

There is potential for the therapeutic targeting of miRNAs in OA. MiRNA expression can be modulated to restore cartilage homeostasis. A synthetic miRNA mimetic or inhibitor, for example, could normalize miRNA profiles in diseased joints. Combining miRNA therapy with other treatments, like mesenchymal stem cell therapy, could enhance therapeutic outcomes. However, challenges such as delivery specificity, miRNA stability, and potential adverse effects must be addressed. Further research is essential to develop effective miRNA delivery methods and understand miRNA/gene interactions in OA.

## 6. Conclusions

In articular cartilage degeneration, enzymes from the metalloproteinase family play a critical role, whose activity is regulated by pro-inflammatory cytokines, transcription factors, and miRNAs. MiRNAs are small, non-structural RNA molecules that regulate gene expression post-transcriptionally. The miRNAs affect protein synthesis by binding to complementary sequences in mRNAs, thereby causing degradation or blocking translation. In the context of osteoarthritis (OA), miRNAs are considered important regulators of anabolic and catabolic processes in articular cartilage, as well as mediators in the inflammatory response and degenerative processes.

Furthermore, miRNAs may prove useful as diagnostic biomarkers in osteoarthritis. Despite many studies suggesting that miRNAs play an important role in OA pathogenesis, some results remain controversial. Different test results can be influenced primarily by the type of sample used: cartilage, synovial fluid, and blood may affect miRNA expression profiles [161]. As well, the complexity of OA resulting from the interaction of many factors such as genetics, age, gender, lifestyle, and overall health may lead to differences in miRNA expression. Studies may also use different research techniques to measure miRNA expression, which may result in different results. Last but not least, statistical analysis methods used in different studies may influence the interpretation of the results.

It is important to remember that treating OA will likely require a multipronged approach that addresses the genetic, biological, and environmental factors contributing to the disease. We may be able to use miRNAs as part of our treatment arsenal against OA in this context.

Thus, miRNAs are critical in the pathophysiology of OA, influencing the processes of degradation and repair of the cartilage. It may be possible to develop new therapeutic and diagnostic strategies by understanding their function. To improve patient outcomes, miRNA therapies may be able to synergize with existing osteoporosis treatments, such as bisphosphonates and monoclonal antibodies. There is, however, still much work to be done to better understand the complex interactions between miRNAs and their target genes. Specifically, further research is needed to determine how miRNA therapies can be delivered to joints, how to ensure target specificity, and how to avoid potential side effects. To mitigate the possibility of adverse effects of miRNA-based therapies, comprehensive studies are needed on their long-term effects and safety.

## Figures and Tables

**Figure 1 ijms-25-06352-f001:**
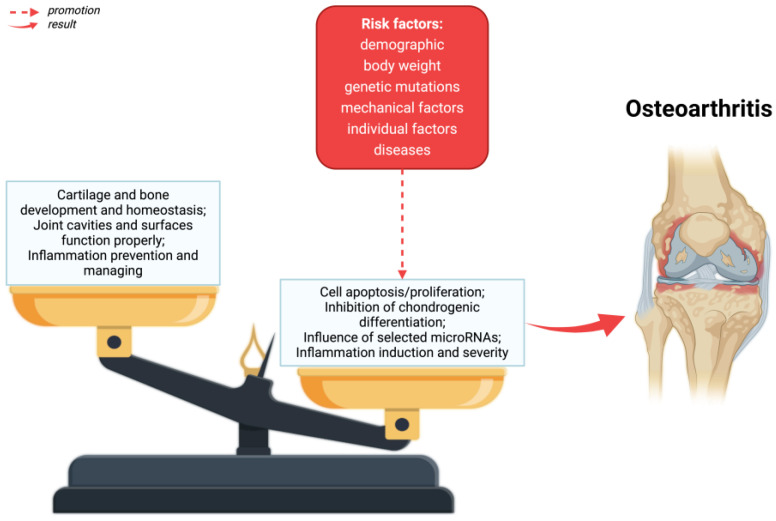
The pathophysiology of osteoarthritis. As a result of risk factors, alterations in homeostasis in the formation and destruction of cartilage and bones, as well as disorders of the joint cavity and joint surfaces can take place. Created with BioRender.com (accessed on 4 June 2024).

**Figure 2 ijms-25-06352-f002:**
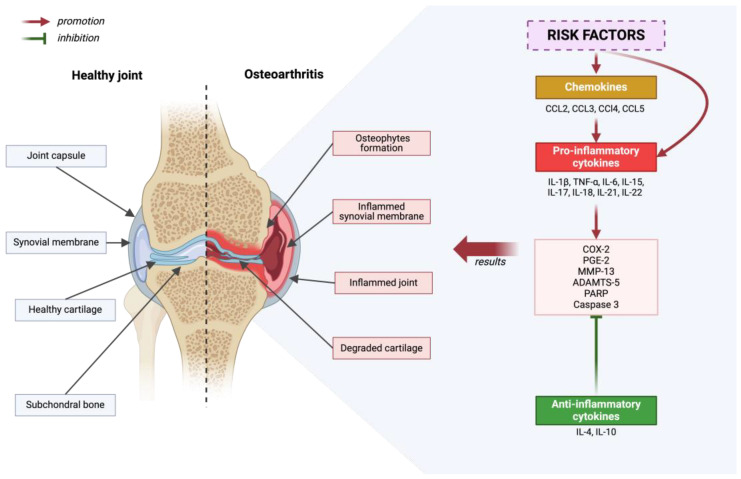
Diagram depicting a healthy joint in comparison to a joint that has been affected by degenerative disease. Inflammatory processes are outlined, considering the promoting and inhibiting effects of molecular factors. Created with BioRender.com (accessed on 4 June 2024).

**Table 1 ijms-25-06352-t001:** Factors increasing osteoarthritis risk.

Group of Risk Factors	Risk Factors	References
demographic	female; older age	[13,15,16,17,18,19,20,21,22,23]
body weight	overweight and obesity	[13,14,15,16,17,18,19]
genetic mutations	e.g., mutation of the *COL2A1*, *COL11A*, *COL11A2*, *COL1A1*, and *COL9A1* gene	[17,21,29,30]
mechanical factors	professional work requiring frequent knee bending and significant use of manual dexterity; practicing competitive sports in the past; weakness of periarticular skeletal muscles; sedentary lifestyle; intense recreational running; past injuries; previous knee surgery	[16,17,18,21,22,24,25,26,27,28,29]
individual factors	high bone mineral density	[27,31,32,33]
diseases	disturbances of deep sensation	[24]

**Table 2 ijms-25-06352-t002:** Classification of secondary osteoarthritis according to the American College of Rheumatology (ACR) [24,36,37].

Categories	Factors
developmental and congenital defects	
local diseases	aseptic necrosis of the femoral head in children; congenital hip dysplasia; exfoliation of the bone epiphysis
mechanical factors	difference in the length of the lower limbs; valgus or varus; joint hypermobility syndrome
dieseases	
metabolic	ochronosis; hereditary hemochromatosis; Wilson’s disease; Gaucher’s disease
endocrine	acromegaly; hyperparathyroidism; diabetes; obesity; hypothyroidism
from the deposition of calcium salts	chondrocalcinosis; apatite arthropathy
endemic diseases	Kashin/Beck disease; Mseleni disease
other bone and joint diseases	local: fractures; aseptic necrosis; infections; gout
	disseminated: rheumatoid arthritis; Paget’s disease; osteopeyrosis; osteochondritis; other inflammations
	neurodystrophy of bones and joint
other diseases	hemoglobinopathies; caisson disease
other factors	
injuries	acute; chronic
external factors	frostbite

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
