# Peer review of "The Role of MicroRNAs in the Pathophysiology of Osteoarthritis"

_ijms, 2024, doi:10.3390/ijms25126352_

Round 1
Reviewer 1 Report
Comments and Suggestions for Authors
The manuscript entitled “The role of MicroRNAs in the Pathophysiology of Osteoarthritis”. It is a review article.
Below are some suggestions:
In the Abstract:
I suggest that the authors write the summary again, as it contains information that practically already exists in the literature. They must include the objective of the review, which was not clear, as well as its clinical importance and a conclusion.
In the Introduction:
- The introduction is very succinct. I suggest a division into topics, such as: 1.1: osteoarthritis; 1.2: risk factors (also providing a written description of the data presented in the table); 1.3 clinical relevance.
- I suggest authors insert the objective of the review in the last paragraph.
2. Pathophysiology of osteoarthritis
- The authors should insert an image for better understanding and illustration of the pathophysiology.
3. Potential therapeutic targets: adjust the item number, wouldn't it be 4?
* Even though it is a review article, it would be important for the authors to insert a "Discussion" item, bringing the results of the main selected articles and making a comparison of the data presented.
5.Conclusions
- I suggest to the authors a more concise conclusion, including applicability and clinical relevance.
Comments on the Quality of English LanguageModerate editing.
Author Response
Dear Editors,
Dear Reviewers,
Enclosed please find our revised review article entitled The role of MicroRNAs in the Pathophysiology of Osteoarthritis by Dariusz Szala, Marta Kopańska, Julia Trojniak, Jarosław Jabłoński, Dorota Hanf-Osetek, Sławomir Snela, Izabela Zawlik.
We improved our manuscript with respect to The Reviewers comments. Below, please find our point by point responses. I believe that we have answered all the comments adequately and have revised the manuscript properly, producing a more balanced and better account of our work.
Yours sincerely,
Marta Kopańska
Reviewer
The manuscript entitled “The role of MicroRNAs in the Pathophysiology of Osteoarthritis”. It is a review article. Below are some suggestions:
In the Abstract:
- I suggest that the authors write the summary again, as it contains information that practically already exists in the literature. They must include the objective of the review, which was not clear, as well as its clinical importance and a conclusion.
Authors Comment: Dear Reviewer, the authors are grateful for your consideration. This review was summarized in a new abstract with a clear focus on the purpose of the review and an emphasis on its clinical significance and conclusions.
In the Introduction:
- The introduction is very succinct. I suggest a division into topics, such as: 1.1: osteoarthritis; 1.2: risk factors (also providing a written description of the data presented in the table); 1.3 clinical relevance.
- I suggest authors insert the objective of the review in the last paragraph.
Authors Comment: Dear Reviewer, we appreciate your valuable comments. Taking your suggestions into consideration, the introduction has been divided into three sections: "1.1: osteoarthritis"; "1.2: risk factors"; and "1.3 clinical relevance". Moreover, each section has been expanded to reflect the most recent publications in the field. In addition, the purpose of the review has been clearly outlined. Hopefully, the changes introduced will result in a better organization of the article.
- Pathophysiology of osteoarthritis
- The authors should insert an image for better understanding and illustration of the pathophysiology.
Authors Comment: Dear Reviewer, you make a very valid point. The pathophysiology of osteoarthritis has been illustrated by an image.
- Potential therapeutic targets: adjust the item number, wouldn't it be 4?
Authors Comment: Dear Reviewer, thank you for bringing this to our attention. The numbering has been corrected.
* Even though it is a review article, it would be important for the authors to insert a "Discussion" item, bringing the results of the main selected articles and making a comparison of the data presented.
Authors Comment: Dear Reviewer, thank you so much for your suggestion. According to the authors, it is absolutely accurate. The "Discussion" (5) chapter was added, in which it collected and compared the most important reports and research results that needed to be highlighted from the references listed.
- Conclusions
- I suggest to the authors a more concise conclusion, including applicability and clinical relevance.
Authors Comment: Dear Reviewer, the conclusions have been modified to emphasize the applicability and clinical relevance of miRNAs.
Reviewer 2 Report
Comments and Suggestions for Authors
This instrumental review article warrants publication. Some suggestions for improvement are outlined below:
1. The format of Table 2 should be reorganised.
2. Line 71: Add the following references in addition to references 2, 24, and 35
https://www.mdpi.com/2227-9059/11/4/1189
https://www.mdpi.com/2075-1729/14/2/269
3. Line 78: Mention that miRNA could also modulate OASF (OA synovial fibroblasts), not only affect chondrocytes.
4. Lines 122-127: The authors describe the role of subchondral bone in OA. Please supplement this paragraph with information from Section 3.3 of the following reference:
https://www.mdpi.com/2227-9059/12/4/843
5. Table 3 should be reorganised. The title "Level miR in OA" could be revised to "Level in OA", and "downregulation/upregulation" could be replaced with ↓/↑. "Attitude towards osteoarthritis" could be revised to "Role in OA pathogenesis", and ↓/↑ could be replaced with +/-.
6. Regarding the diagnostic role of miRNA in OA (lines 345-360), this paragraph should be shifted to the main text. Avoid including excessive information in the conclusion section that has not been mentioned in the main text.
Comments on the Quality of English LanguageMinor editing of English language required
Author Response
Reviewer 2
This instrumental review article warrants publication. Some suggestions for improvement are outlined below:
- The format of Table 2 should be reorganised.
Authors Comment: Dear Reviewer, thank you for your valid comment. The format of Table 2 has been reorganized. It is hoped that the table will now be more understandable.
- Line 71: Add the following references in addition to references 2, 24, and 35
https://www.mdpi.com/2227-9059/11/4/1189
https://www.mdpi.com/2075-1729/14/2/269
Authors Comment: Dear Reviewer, it is greatly appreciated that you recommended these valuable articles. The references have been added.
- Line 78: Mention that miRNA could also modulate OASF (OA synovial fibroblasts), not only affect chondrocytes.
Authors Comment: Dear Reviewer, the suggestion you have provided is greatly appreciated. In accordance with your instructions, the information has been updated.
- Lines 122-127: The authors describe the role of subchondral bone in OA. Please supplement this paragraph with information from Section 3.3 of the following reference:
https://www.mdpi.com/2227-9059/12/4/843
Authors Comment: Dear Reviewer, thank you very much for your suggestion. A supplement has been added to the paragraph as suggested. Information was added from the suggested reference and expanded from other sources.
- Table 3 should be reorganised. The title "Level miR in OA" could be revised to "Level in OA", and "downregulation/upregulation" could be replaced with ↓/↑. "Attitude towards osteoarthritis" could be revised to "Role in OA pathogenesis", and ↓/↑ could be replaced with +/-.
Authors Comment: Dear Reviewer, thank you very much for presenting these suggestions. The table 3 has been reorganized in accordance with your recommendations. The title " Level miR in OA" was changed to "Level in OA" and "downregulation/upregulation" was replaced by ↓/↑. "Attitude towards osteoarthritis" was changed to "Role in OA pathogenesis" and ↓/↑ was replaced by +/-.
- Regarding the diagnostic role of miRNA in OA (lines 345-360), this paragraph should be shifted to the main text. Avoid including excessive information in the conclusion section that has not been mentioned in the main text.
Authors Comment: Dear Reviewer, thank you very much for your comment. In the main text, lines 347-360 have been moved to chapter 3 “The role of miRNAs in the pathogenesis of osteoarthritis”, lines 347-352 and chapter 4 "Potential therapeutic targets", lines 374-380.
Reviewer 3 Report
Comments and Suggestions for Authors
The article The Role of MicroRNAs in the Pathophysiology of Osteoarthritis explores the critical role of microRNAs (miRNAs) in osteoarthritis (OA), a leading cause of joint pain in the elderly. It highlights miRNAs as regulators of gene expression and their involvement in inflammatory processes, cartilage metabolism, and cell apoptosis, all of which contribute to OA progression. The article also suggests the potential of miRNAs as biomarkers and therapeutic targets, offering new possibilities for early diagnosis and innovative treatments. Overall, the article provides valuable insights into the complex mechanisms of OA and the promising role of miRNAs in its management.
The article is interesting and generally well-written, however, it will require minor revisions before further processing and acceptance for publication. I have included detailed comments below.
Minor comments:
Please highlight the purpose of the article and the importance of reviewing the state of knowledge in the context of further research.
The introduction is very short, please add more detailed information on degenerative disease with the necessary recent literature. Please emphasize the importance of the problem under consideration and more clearly state the purpose of this article.
Please emphasize that cartilage damage along with other factors such as work, sports participation, musculoskeletal injuries, obesity and gender can influence the formation and progression of degenerative changes in the joint. Information about this, along with the necessary literature, should be added in the in the 1st paragraph (lines 31-42) of the introduction. Authors may find some useful information in the works: doi: 10.35784/acs-2023-40; DOI: 10.1056/NEJMcp1903768; DOI 10.3390/app10238312; DOI: 10.1056/NEJMcp051726; doi:10.35784/acs-2022-14; https://doi.org/10.1007/s10787-011-0118-0
Please add this information along with the current literature in the introduction by also including it in Table 1.
Table 2 is not very readable in its current form, I recommend rebuilding it or changing the formatting.
In conclusion, The Role of MicroRNAs in the Pathophysiology of Osteoarthritis is a well-structured and informative article that significantly enhances our understanding of the role of miRNAs in OA. The authors successfully integrate current knowledge with emerging research, providing a valuable resource for researchers and clinicians alike. By elucidating the complex interactions between miRNAs and various cellular processes in OA, the article paves the way for future studies aimed at developing novel diagnostic and therapeutic approaches for this debilitating condition.
I congratulate the authors on an interesting study. After making appropriate corrections and additions to the text and literature, the work can be accepted for publication.
Comments on the Quality of English LanguageMinor editing of English language required.
Author Response
Dear Reviewer,
Enclosed please find our revised review article entitled The role of MicroRNAs in the Pathophysiology of Osteoarthritis by Dariusz Szala, Marta Kopańska, Julia Trojniak, Jarosław Jabłoński, Dorota Hanf-Osetek, Sławomir Snela, Izabela Zawlik.
We improved our manuscript with respect to The Reviewers comments. Below, please find our point by point responses. I believe that we have answered all the comments adequately and have revised the manuscript properly, producing a more balanced and better account of our work.
Yours sincerely,
Marta Kopańska
Reviewer
The article The Role of MicroRNAs in the Pathophysiology of Osteoarthritis explores the critical role of microRNAs (miRNAs) in osteoarthritis (OA), a leading cause of joint pain in the elderly. It highlights miRNAs as regulators of gene expression and their involvement in inflammatory processes, cartilage metabolism, and cell apoptosis, all of which contribute to OA progression. The article also suggests the potential of miRNAs as biomarkers and therapeutic targets, offering new possibilities for early diagnosis and innovative treatments. Overall, the article provides valuable insights into the complex mechanisms of OA and the promising role of miRNAs in its management.
The article is interesting and generally well-written, however, it will require minor revisions before further processing and acceptance for publication. I have included detailed comments below.
Minor comments:
- Please highlight the purpose of the article and the importance of reviewing the state of knowledge in the context of further research.
Authors Comment: Dear Reviewer, thank you very much for your attention. A clear definition of the purpose and importance of the review has been added to both the abstract and introduction.
- The introduction is very short, please add more detailed information on degenerative disease with the necessary recent literature. Please emphasize the importance of the problem under consideration and more clearly state the purpose of this article.
Authors Comment: Dear Reviewer, we appreciate your valuable comments. Taking your suggestions into consideration, the introduction has been divided into three sections: "1.1: osteoarthritis"; "1.2: risk factors"; and "1.3 clinical relevance". Moreover, each section has been expanded to reflect the most recent publications in the field. In addition, the purpose of the review has been clearly outlined. Hopefully, the changes introduced will result in a better organization of the article.
- Please emphasize that cartilage damage along with other factors such as work, sports participation, musculoskeletal injuries, obesity and gender can influence the formation and progression of degenerative changes in the joint. Information about this, along with the necessary literature, should be added in the in the 1st paragraph (lines 31-42) of the introduction. Authors may find some useful information in the works: doi: 10.35784/acs-2023-40; DOI: 10.1056/NEJMcp1903768; DOI 10.3390/app10238312; DOI: 10.1056/NEJMcp051726; doi:10.35784/acs-2022-14; https://doi.org/10.1007/s10787-011-0118-0
- Please add this information along with the current literature in the introduction by also including it in Table 1.
Authors Comment: Dear Reviewer, in response to Reviewer 1's suggestion, the introduction has been divided into three sections. In response to your suggestions, the risk factors section has been updated. The valuable sources you suggested have all been included.
- Table 2 is not very readable in its current form, I recommend rebuilding it or changing the formatting.
Authors Comment: Dear Reviewer, thank you for your valid comment. The format of Table 2 has been reorganized. It is hoped that the table will now be more understandable.
Round 2
Reviewer 1 Report
Comments and Suggestions for Authors
I would like to thank the authors of the manuscript for making all the suggestions, further improving the organization and scientific basis for publication.
Comments on the Quality of English LanguageMinor editing.